# App2Check and Tweet2Check: machine learning-based tools for Sentiment Analysis of Apps Reviews and Tweets

## Abstract

Sentiment Analysis has nowadays a crucial role in social media analysis and, more generally, in analyzing user opinions about general topics or user reviews about product/services, enabling a huge number of applications. Many methods and software implementing different approaches exist and there is not a clear best approach for Sentiment classification/quantification. We believe that performance reached by machine learning approaches is a key advantage to apply to sentiment analysis in order to reach a performance which is very close to the one obtained by a group of humans, who evaluate subjective sentences such as user reviews. In this paper, we present the results of our experimental evaluation of both research and commercial state-of-the-art tools for Sentiment Analysis, considering as benchmarks, both user reviews related to the evaluation of apps published to app stores and tweets. Thus, we show that our tools App2Check and Tweet2Check –developed mainly applying supervised learning techniques– are the best tools for sentiment evaluation on apps reviews and tweets for both Italian and English language, compared to the state-of-the-art research tools.

## 1 Introduction

Sentiment Analysis has nowadays a crucial role in social media analysis and, more generally, in analysing user opinions about general topics or user reviews about product/services, enabling a huge number of applications. For instance, sentiment analysis can be applied to monitoring the reputation or opinion of a company or a brand with the analysis of reviews of consumer products or services (Hu and Liu, 2004), marketing campaigns in politics (Tumasjan et al., 2010), and financial applications (Oliveira et al., 2013) (Bollen et al., 2010). App stores can be seen as another, not yet well explored, field of application of sentiment analysis. Indeed, they are another social media where users can freely express their own opinion through app reviews about a product, i.e. the specific app under evaluation, or a service, to which the considered app is connecting the user (e.g., a mobile banking app connects users to mobile banking services). In addition, reading user reviews on app stores shows that people frequently talk about and evaluate also the brand associated to the app under review: thus, it is possible to extract people opinion about a brand or the sentiment about a company or the provided service quality.

Twitter, on the other hand, is one of the most popular and largely used social networks which is very interesting to monitor from the perspective of sentiment analysis, and that is already considered in research challenges in this research area.

In this paper, we focus on the app store as a social media platform and on the sentiment evaluation of app reviews and tweets. The former are examples of reviews related to a product, or a service or the associated brand; the latter can include more general sentences, which can be significantly different from reviews as for app reviews, and can be even more difficult to evaluate respect to sentiment analysis. On the other hand, apps reviews are also interesting because they include a score which is not available for tweets. Moreover, in apps reviews, the sentiment score detected in a comment can significantly differ from the score assigned by the user to the app under evaluation. For example, a user can assign his good score to the app (i.e. assigning 5 stars) but also express in natu-

ral language some suggestions or highlight some minor bugs that do not influence his overall app evaluation. For example, the comment *Excellent app, but it crashes and closes while code scanning of with camera. Do something!* was rated 4 stars by the user, but the sentence contains overall a negative sentiment from the perspective of the app developer since it explains a serious bug. All of this non-structured information is fully missing by only superficially evaluating an app through a 1 to 5 overall score – or any other product evaluated by the user with a sentence and a score–.

About the methods of processing user reviews and tweets, many methods and software implementing different approaches exist and there is not a clear best approach for Sentiment classification/quantification (Araújo et al., 2016) (Chen, 2010) (Gao and Sebastiani, 2015). In our opinion, performance reached by machine learning approaches is a key advantage to apply to sentiment analysis in order to reach a performance which is very close to the one obtained by a group of humans evaluating subjective sentences such as user reviews. As a reference sentiment score, it is well known in literature that a group of humans agree only about in the 80% of cases when evaluating sentiment sentences (Wilson et al., 2009) (Grimes, 2010).

In this paper, we present the App2Check and Tweet2Check tools in their English and Italian versions, built on top of two specialized predictive models developed applying supervised learning techniques, and present the results of our experimental evaluation. This shows that App2Check outperforms 19 state-of-the-art research tools on 11 thousand apps reviews in Italian, by overcoming the theoretical reference of 80% of accuracy; and it is better than those tools on 11 thousand app reviews in English. About apps reviews we considered for experiments, these are new test sets we are making available for the research community. About tweets, we show that Tweet2Check is better than competitors on 3899 tweets in Italian language and on 1 thousand randomly selected tweets in English from the latest competitions.

The structure of the paper is the following. In section 2 we report a brief description of research tools iFeel platform, SentiStrength for Italian and our tools, App2Check and Tweet2Check. In section 3 we present and discuss our experimental evaluation and in section 4 we show the conclusions.

## 2 State-of-the-art Research Tools

In this section we describe our tools and briefly describe iFeel (Araújo et al., 2014) (v.2.0), a research platform which allows to perform experimental evaluation on 19 state-of-the-art research tools for sentiment analysis, SentiStrength for Italian, which we used to perform our experimental evaluation on Italian language –which is currently our main focus– and on English language, and we describe our systems .

### 2.1 App2Check/Tweet2Check description

App2Check and Tweet2Check implement supervised learning techniques that allowed us to create two predictive models for sentiment quantification: one specialized on apps reviews and one on tweets. Training of predictive models is performed by considering a huge variety of language domains and different kinds of user reviews. Both tools provide, as answer to a sentence in Italian or English language, a quantification of the sentiment polarity scored from 1 to 5, according to the most recent trend shown in the last sentiment evaluation SemEval (Sem, 2016), where tracks considering quantification have been introduced. Thus, we consider the following quantification: as positive, sentences with score 4 (positive) or 5 (very positive); as negative, sentences with score 1 (very negative) or 2 (negative); as neutral, sentences with score 3. In order to compute the final answer, App2Check/Tweet2Check also apply a set of algorithms which take into account some natural language processing techniques, allowing e.g. to also automatically perform topic/named entity extraction. It is not possible to give more details on the engine due to non-disclosure restrictions.

App2Check and Tweet2Check are not only constituted by a web service providing access to the sentiment prediction of sentences (with free trial for research purposes), but also provide a full user-friendly web application whose features are out of the scope of this paper. A full demo of the tools will be available for paper presentation.

### 2.2 iFeel Platform

iFeel is a research web platform (Araújo et al., 2016; Araújo et al., 2014) allowing to run multiple sentiment analysis tools on the specified list of sentences. It allows to natively run tools support-

ing English and to first translate sentences from other languages to English and then run 19 tools on the English translated sentences. Since it has been experimentally shown in (Araújo et al., 2016) that well known language specific methods do not have a significant advantage over a simple machine translation approach before sentiment evaluation, and since most of the research tools do not have a publicly available Italian version, the results of the tools ran by iFeel are used for both Italian and English. However, we also considered for the comparison the only research tool we found which natively processes Italian language (Sentistrenght) and it is our reference tool with no translation before sentiment evaluation.

The tools included in iFeel are (in alphabetical order): AFINN, Emolex, Emoticon Distance Supervised, Emoticons, Happiness Index, NRC Hashtag, Opinion Finder, Opinion Lexicon, Panas-t, SANN, SASA, Senticnet, Sentiment140, SentiStrength, SentiWordNet, SO-CAL, Stanford Deep Learning, Umigon, Vader.

### 2.3 Sentistrength for Italian

SentiStrength was produced as part of the Cyber-Emotions project, supported by EU FP7. It estimates the strength of positive and negative sentiment in short texts, even for informal language. According to the authors, it has human-level accuracy for short social web texts in English, except political texts (Thelwall et al., 2010). SentiStrength authors make available a version of the tool which natively manages Italian language. All tests have been carried out with both average emotion and strongest emotion options, but in this paper we only report the results obtained with the latter option turned on, due to better performance. Since the English version of SentiStrength is also included in iFeel, in the following we call the Italian version SentiStrengthIta.

## 3 Experimental Evaluation

In our experimental evaluation we evaluated App2Check and Tweet2Check sentiment predictors on user reviews of apps from Apple App Store and Google Play Store and on tweets, for both Italian and English language.

Apps reviews benchmark set is constituted by: *i*. *Test sets A-ita* and *A-eng* , both constituted by 1 thousand comments from the famous Candy Crush Saga app, respectively in Italian and En-

glish. We performed a manual quantification (in the 1-5 range) of the sentiment, performed by a trained person, in order to have a reference sentiment score (from now on called human sentiment quantification, or HSQ). *ii*. *Test set B* made of 10 thousands comments from the following 10 different very popular apps (one thousand comments per app).

- *Test set B-ita*: Angry Birds, Banco Posta, Facebook, Fruit Ninja, Gmail, Mobile Banking Unicredit, My Vodafone, PayPal, Twitter, Whatsapp.

- *B-eng*: Candy Crush Soda Saga, Chase Mobile, Clash Of Clans, Facebook Messenger, Gmail, Instagram, My Verizon, PayPal, Snapchat and Wells Fargo.

Every comment has a score, called app rating, associated to the app.

Tweet benchmark set is constituted by 3899 tweets in Italian language from (Araújo et al., 2016) (and (Sem, 2016)) and 1000 tweets in English randomly selected from the English dataset from the same source. While the latter does not contain the neutral class, the former contains also neutral sentences.

All tables show columns macro F1 (MF1), accuracy (Acc), F1 on each class (resp. F1(-) for negative, F1(x) for neutral –if present–, and F1(+) for positive), and are sorted by macro F1, considered as scoring system for tools. In all of the tables We highlight in bold the winner tool per column. We ran App2Check/Tweet2Check, the 19 research tools through iFeel on all test sets, and SentiStrengthIta on *Test set A-ita*, *Test set B-ita* and Italian tweets, in order to add a comparison with a tool supporting Italian.

All of the benchmarks are submitted with the paper, together with a demo access to App2Check and Tweet2Check prediction web services, in order to extend or make repeatable the experiments.

### 3.1 Evaluation on App reviews

Results shown in Table 1 highlight that App2Check outperforms all of the research tools in Italian with respect to HSQ, overcoming the theoretical 80% of accuracy; in Table 2 is shown that our tool is better than the research ones also for English apps reviews. Very good results are obtained in both languages by Sentiment 140, which has the best macro-F1 and accuracy among

| Tool | M-F1 | Acc | F1(-) | F1(x) | F1(+) |
|---|---|---|---|---|---|
| App2Check | **65.8** | **81.8** | 85.9 | 25.2 | **86.4** |
| Sentiment140 | 58.1 | 71.6 | 80.4 | 21.4 | 72.5 |
| SentiWordNet | 57.2 | 67.3 | 73.5 | **26.6** | 71.4 |
| Stanford DL | 53.7 | 60.5 | 70.0 | 21.2 | 69.9 |
| NRC Hashtag | 52.9 | 81.0 | 76.4 | 16.4 | 66.0 |
| Umigon | 50.9 | 56.4 | 54.0 | 20.5 | 78.1 |
| Sentistrength | 50.4 | 57.8 | 47.5 | 25.6 | 78.0 |
| Op. Lexicon | 49.8 | 54.2 | 53.1 | 23.7 | 72.6 |
| AFINN | 49.7 | 57.4 | 52.4 | 21.8 | 74.9 |
| Vader | 40.9 | 42.8 | 31.9 | 22.8 | 68.1 |
| Senticnet | 40.2 | 50.6 | 37.4 | 19.7 | 63.6 |
| Emolex | 40.0 | 43.3 | 43.1 | 16.1 | 60.9 |
| SASA | 39.3 | 44.3 | 40.8 | 15.8 | 61.3 |
| SO-CAL | 39.0 | 40.8 | 45.2 | 17.0 | 54.8 |
| SentiStr. Ita | 38.2 | 41.5 | 34.3 | 20.5 | 59.7 |
| H. Index | 32.3 | 39.8 | 16.9 | 18.6 | 61.4 |
| Op. Finder | 23.6 | 22.9 | 24.9 | 18.1 | 27.9 |
| Emoticon DS | 21.5 | 41.6 | 1.8 | 4.1 | 58.7 |
| SANN | 16.5 | 17.8 | 1.8 | 19.9 | 27.7 |
| Panas-t | 7.4 | 11.0 | 1.3 | 18.7 | 2.2 |
| Emoticons | - | 10.3 | - | 18.6 | - |

Table 1: Comparison on 1K Candy Crush Saga app reviews in Italian wrt HSQ

| Tool | M-F1 | Acc | F1(-) | F1(x) | F1(+) |
|---|---|---|---|---|---|
| App2Check | **61.9** | **74.5** | 82.0 | 23.2 | **80.5** |
| Sentiment140 | 59.9 | 70.1 | 77.2 | 29.7 | 73.0 |
| Stanford DL | 57.6 | 61.9 | 71.2 | 33.3 | 68.3 |
| NRC Hashtag | 56.8 | 64.9 | 75.6 | 31.7 | 63.2 |
| Umigon | 56.4 | 59.9 | 59.1 | 33.3 | 76.8 |
| Op. Lexicon | 53.0 | 55.6 | 54.7 | **36.5** | 67.7 |
| AFINN | 52.8 | 57.6 | 54.1 | 33.6 | 70.9 |
| SentiWordNet | 52.5 | 58.9 | 63.4 | 31.5 | 62.5 |
| SentiStrength | 46.3 | 49.9 | 44.0 | 29.1 | 65.9 |
| SO-CAL | 43.1 | 44.5 | 50.4 | 27.2 | 51.6 |
| Emolex | 41.3 | 43.1 | 38.0 | 32.6 | 53.4 |
| Vader | 40.4 | 43.0 | 24.8 | 31.8 | 64.7 |
| SASA | 40.0 | 44.4 | 40.0 | 19.2 | 60.7 |
| Senticnet | 38.1 | 43.0 | 32.9 | 29.0 | 52.4 |
| H. Index | 33.9 | 40.1 | 19.7 | 26.4 | 55.7 |
| Op. Finder | 27.9 | 66.9 | 29.5 | 26.3 | 28.0 |
| SANN | 19.2 | 22.8 | 0.9 | 29.9 | 26.8 |
| Panas-t | 11.1 | 17.7 | 3.6 | 29.1 | 0.5 |
| Emoticon DS | - | 39.9 | - | 15.8 | 56.2 |
| Emoticons | - | 16.8 | - | 28.8 | - |

Table 2: Comparison on 1K Candy Crush Saga app reviews in English wrt HSQ

research tools, and is closer to App2Check on *Test set A-eng*. We can see that for all of the tools is hard to identify neutral reviews, according to the low F1(x) score, despite a good performance on positive and negative class.

We experienced that, as already said, considering a single comment, the score/rating expressed by a user respect to an app can be in general substantially different respect to the sentiment expressed by a human. However, the average score/rating of many (hundreds of) comments can

be an approximation of the average sentiment expressed by a human on the same set. Indeed, we compared HSQ with Candy Crush Saga app rating and they agree on about 80% of cases: this data allows us to consider app rating as an approximated indicator of sentiment when averaging thousands of comments.

| Tool | M-F1 | Acc | F1(-) | F1(x) | F1(+) |
|---|---|---|---|---|---|
| **App2Check** | **73.3** | **85.7** | **82.7** | **45.6** | **91.7** |
| SentiWordNet | 47.9 | 65.9 | 60.4 | 6.2 | 77.1 |
| AFINN | 47.5 | 60.3 | 49.2 | 16.6 | 76.7 |
| SentiStrength | 47.5 | 59.7 | 46.3 | 19.3 | 76.8 |
| Stanford DL | 45.6 | 54.0 | 56.5 | 13.5 | 66.8 |
| Op. Lexicon | 44.9 | 55.3 | 45.1 | 17.5 | 72.2 |
| Sentiment140 | 44.1 | 58.7 | 57.4 | 6.7 | 68.2 |
| Umigon | 42.8 | 50.1 | 47.8 | 14.6 | 66.2 |
| SO-CAL | 41.8 | 49.3 | 45.8 | 13.8 | 65.6 |
| NRC Hashtag | 41.2 | 65.7 | 53.6 | 8.3 | 61.7 |
| Senticnet | 40.9 | 63.1 | 36.6 | 9.1 | 76.9 |
| Vader | 38.5 | 46.2 | 29.5 | 19.7 | 66.3 |
| Emolex | 38.3 | 45.5 | 38.5 | 14.1 | 62.3 |
| SASA | 37.7 | 48.8 | 29.6 | 16.4 | 67.1 |
| SentiStr. Ita | 34.0 | 39.6 | 32.0 | 13.9 | 56.2 |
| H. Index | 31.1 | 39.0 | 21.9 | 13.4 | 57.9 |
| Emoticon DS | 27.8 | 63.6 | 3.0 | 2.5 | 77.8 |
| Op. Finder | 26.0 | 26.6 | 25.3 | 15.4 | 37.2 |
| SANN | 12.2 | 14.9 | 3.6 | 16.5 | 16.4 |
| Panas-t | 6.0 | 9.0 | 1.2 | 16.0 | 0.9 |
| Emoticons | 5.4 | 8.8 | 0.1 | 15.9 | 0.4 |

Table 3: Comparison on 10.000 apps reviews in Italian wrt app rating.

| Tool | M-F1 | Acc | F1(-) | F1(x) | F1(+) |
|---|---|---|---|---|---|
| App2Check | **53.4** | **61.2** | **66.1** | 22.5 | **71.7** |
| Umigon | 48.8 | 51.1 | 52.7 | 29.4 | 64.4 |
| Stanford DL | 47.5 | 51.4 | 60.7 | 24.2 | 57.4 |
| AFINN | 45.5 | 49.9 | 46.0 | 27.3 | 63.4 |
| Op. Lexicon | 45.0 | 48.1 | 43.4 | 28.5 | 63.2 |
| SentiStrength | 45.0 | 47.5 | 42.9 | 30.1 | 62.0 |
| SO-CAL | 42.4 | 44.7 | 46.5 | 25.0 | 55.7 |
| Sentiment140 | 42.0 | 53.7 | 64.3 | 9.2 | 52.5 |
| Vader | 40.5 | 42.1 | 26.8 | **34.3** | 60.4 |
| NRC Hashtag | 40.5 | 51.8 | 63.4 | 9.2 | 48.9 |
| SentiWordNet | 40.3 | 50.8 | 52.9 | 8.0 | 60.0 |
| Emolex | 39.3 | 40.7 | 37.9 | 27.6 | 52.3 |
| SASA | 38.0 | 40.2 | 34.8 | 26.4 | 52.7 |
| H. Index | 34.4 | 37.7 | 22.2 | 28.3 | 52.7 |
| Senticnet | 33.5 | 44.1 | 32.7 | 10.7 | 57.1 |
| Op. Finder | 33.1 | 32.9 | 30.6 | 29.9 | 38.7 |
| Emoticon DS | 20.1 | 39.6 | 1.5 | 2.0 | 56.7 |
| SANN | 17.2 | 23.5 | 1.5 | 34.0 | 16.1 |
| Panas-t | 12.1 | 20.4 | 2.1 | 33.2 | 1.1 |
| Emoticons | - | 20.0 | - | 33.4 | 0.1 |

Table 4: Comparison on 10.000 apps reviews in English wrt app rating.

In Tables 3 and 4 we extend our evaluation respectively on *Test set B-ita* and *Test set B-eng* with respect to *app rating*, in order to have a reference

indicator when HSQ is not available. We can see that App2Check generalizes well when tested on many 10 thousand reviews and is the best tool for all measures on the Italian test set, outperforming the research tools with about a 20% of better accuracy from the second tool in the table, and shows also a 45% of better accuracy respect to SentiStrenght for Italian. App2Check is also the best tool for the English app reviews on all but the F1(x) measure.

On these test sets, it is difficult to identify the best research tool or the tool having results that are similar to the ones obtained by App2Check. Also Sentiment140, which has been the second best tool (and the best among the research ones) on Candy Crush Saga reviews, is the seventh and the eighth on, respectively, *Test set B-ita* and *Test set B-eng*, even if it still has the highest accuracy among the research tools. We can see that its low macro-F1 is due to the bad results on neutral sentences, which, as previously noticed, are the most difficult to identify.

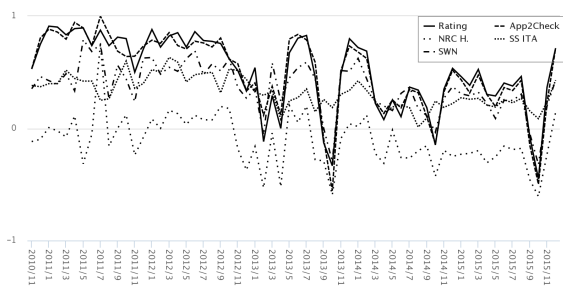

Figure 1: Sentiment quantification plot on 10 thousand app reviews in Italian language.

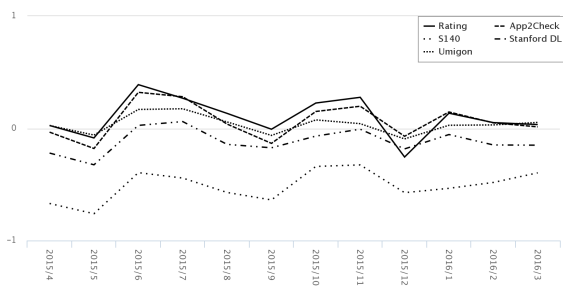

Figure 2: Sentiment quantification plot on 10 thousand app reviews in English language.

In order to understand the reason why App2Check is so much better than the other tools on apps reviews, we show in Figures 1 and 2 two plots of the average sentiment per month of the comments, which have been chronologically sorted. In order to obtain a better representation of the plot, only the most significant research tools are plotted together with Rating and App2Check.

In Figure 1, we choose SentiWordNet (SWN in the plot), which has the best Macro-F1 on this test set, NRC Hashtag (NRC H.), which has the best Accuracy, and SentiStrengthIta (SS. ITA), which is the only research tool supporting Italian. We can see that all tools have similar trend, but NRC Hashtag plot is widely shifted down, and the points in the plots of both SentiWordNet and SentiStrengthIta are closer to the global average sentiment. App2Check plot, instead, is almost overlapped to the rating plot.

In Figure 2, we choose Umigon and Stanford Deep Learning (Stanford DL), which have the best Macro-F1, and Sentiment140 (S140), which has the best accuracy, despite its low Macro-F1. In order to have a clearer chart, we show only the last 12 months. First of all, we can see that Sentiment140 plot highlights a tendency to classify sentences as negative. Once again we can see that all tools follows the trend of the rating, but Stanford Deep Learning is, except for 2015/12, on average more negative than the rating, and Umigon has a plot closer to the rating among the research tools. App2Check is again the closest to the app rating.

We think that App2Check on *Test set B* has even higher accuracy, considering as a reference the HSQ instead of the app rating: this is made clear while using the web application and evaluating the system answers on every single user comment.

## 3.2 Evaluation on Tweets

The first highlighted rows (with *) in Tables 5, 6 and 7 show our tools feature –native thanks to the machine learning approach–, allowing to learn and extend our model with the provided examples. This means that the predicted model has been updated including in the training set also the sentences belonging to the specific benchmarks under evaluation. We show such results for tweets since, compared to app reviews, all of the tools show a lower accuracy which is always under the theoretical 80%. In this cases, having a tool allowing a training on a specific domain, may in general help to reach a higher accuracy: this is useful when a client asks for a high accuracy on a specific domain, e.g. politics, movies, etc. In this cases, our tool –and in general– machine learning-based approaches– allow to meet this important goal by

performing learning on a subset of sentences of the required domain, while different approaches with no learning feature cannot reach it or anyway reach it with more effort.

| Tool | MF1 | Acc | F1(-) | F1(x) | F1(+) |
|------|-----|-----|-------|-------|-------|
| **Tweet2Check\*** | **88.2** | **88.7** | **89.0** | **89.8** | 85.9 |
| Tweet2Check | **46.0** | 46.4 | **52.1** | 39.3 | **46.6** |
| SentiStr. Ita | 44.7 | **46.7** | 39.4 | 54.3 | 40.5 |
| SentiStrength | 41.4 | 43.1 | 38.8 | 52.9 | 32.5 |
| SO-CAL | 40.4 | 42.7 | 37.9 | 54.1 | 29.3 |
| AFINN | 40.4 | 42.2 | 36.3 | 54.4 | 30.5 |
| Umigon | 40.1 | 46.3 | 30.4 | **61.7** | 28.2 |
| Op. Lexicon | 39.0 | 42.9 | 32.9 | 56.6 | 27.6 |
| Op. Finder | 37.0 | 44.9 | 27.7 | 60.1 | 23.0 |
| Vader | 36.1 | 44.0 | 21.4 | 60.1 | 26.9 |
| Emolex | 36.0 | 37.3 | 39.6 | 42.1 | 26.2 |
| SASA | 35.7 | 37.2 | 30.8 | 48.1 | 28.1 |
| Stanford DL | 35.0 | 38.3 | 44.3 | 38.3 | 22.4 |
| H. Index | 33.0 | 36.3 | 20.0 | 49.0 | 29.9 |
| SentiWordNet | 32.0 | 32.9 | 40.3 | 24.4 | 31.2 |
| Sentiment140 | 29.3 | 33.5 | 49.3 | 12.3 | 26.4 |
| NRC Hashtag | 28.3 | 35.1 | 50.5 | 11.8 | 22.6 |
| Senticnet | 25.1 | 26.8 | 28.2 | 14.8 | 32.2 |
| Emoticons | 22.5 | 42.9 | 1.0 | 60.0 | 6.6 |
| Panas-t | 22.1 | 42.6 | 1.6 | 59.7 | 5.0 |
| SANN | 20.4 | 42.7 | 0.7 | 59.8 | 0.7 |
| Emoticon DS | 12.1 | 21.0 | 0.6 | 1.4 | 34.4 |

Table 5: Comparison on 3899 tweets in Italian

Evaluating tools on tweets in Italian language, we can see in Table 5 that Tweet2Check is better than research tools with respect to M-F1, but in this case tools show a closer performance, and the Italian version of SentiStrength has almost the same (slightly higher) accuracy.

In table 6 we show the results on the same test set in a different way, i.e. excluding the tweets that have been manually classified as neutral, since this helps to clear the influence of the neutral class on the performance of all systems. Tweets, indeed, can be as said in general very different than app reviews, and tweets manually labeled as neutral may not contain any opinion and being objective sentences; these sentences can be more difficult to recognize as neutral by tools.

We can see that the MF1 ranking changed, but Tweet2Check is still the best tool, and obtained a Macro-F1 that is 12% higher than Sentiment140, which, once again, is the second best tool.

Finally, in Table 7, results for English tweets are reported, and we can see that, considering only the macro-F1 measure, three tools, Tweet2Check, AFINN, and SentiStrength outperform the others and obtained similar scores, but, considering accuracy too, Tweet2Check is the best one, since it reaches a score of 78.4%, while AFINN, which is

| Tool | MF1 | Acc | F1(-) | F1(+) |
|------|-----|-----|-------|-------|
| **Tweet2Check\*** | **90.5** | **86.7** | **91.4** | **89.7** |
| Tweet2Check | **63.3** | **57.8** | 66.9 | **59.7** |
| Sentiment140 | 51.1 | 53.1 | 63.8 | 38.4 |
| NRC Hashtag | 49.5 | 56.0 | **69.5** | 29.5 |
| SentiStr. Ita | 47.7 | 36.5 | 45.5 | 50.0 |
| SentiWordNet | 47.2 | 46.0 | 50.5 | 43.9 |
| Alchemy | 45.9 | 34.9 | 36.5 | 55.3 |
| Semantria | 45.2 | 33.0 | 40.7 | 49.7 |
| SentiStrength | 43.3 | 36.9 | 46.8 | 39.7 |
| Stanford DL | 42.8 | 43.1 | 60.3 | 25.3 |
| Emolex | 41.8 | 36.4 | 49.7 | 34.0 |
| SO-CAL | 40.4 | 33.5 | 43.5 | 37.4 |
| AFINN | 40.3 | 34.2 | 42.1 | 38.5 |
| Senticnet | 40.3 | 40.3 | 31.3 | 49.3 |
| SASA | 36.5 | 29.8 | 34.8 | 38.1 |
| Op. Lexicon | 35.7 | 27.9 | 37.4 | 33.9 |
| Umigon | 32.9 | 24.4 | 33.4 | 32.4 |
| H. Index | 31.3 | 24.0 | 21.9 | 40.7 |
| Op. Finder | 28.3 | 19.9 | 30.6 | 26.1 |
| Vader | 27.1 | 18.5 | 22.9 | 31.4 |
| Emoticon DS | 26.8 | 36.0 | 0.6 | 53.1 |
| Emoticons | 3.9 | 1.7 | 1.0 | 6.7 |
| Panas-t | 3.4 | 1.5 | 1.7 | 5.1 |
| SANN | 0.7 | 0.4 | 0.7 | 0.7 |

Table 6: Comparison on Italian tweets, without neutral class

the second best tool, only reached 70.9%.

## 4 Conclusion

In this paper we presented our models for sentiment analysis, App2Check and Tweet2Check, in their Italian and English versions, based on two predictive models for apps reviews and tweets applying supervised learning techniques. We showed that App2Check outperforms research competitors on apps reviews, and that Tweet2Check is better than competitors on tweets in both Italian and English language.

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

| Tool | MF1 | Acc | F1(-) | F1(+) |
|---|---|---|---|---|
| **Tweet2Check*** | **85.0** | **84.1** | **84.7** | **85.3** |
| Tweet2Check | **79.1** | **78.4** | 78.2 | **80.0** |
| AFINN | 77.2 | 70.9 | **79.6** | 74.8 |
| Sentistrength | 75.1 | 66.8 | 76.3 | 74.0 |
| Senticnet | 68.4 | 66.0 | 64.2 | 72.5 |
| Umigon | 68.0 | 58.8 | 69.2 | 66.7 |
| Vader | 67.1 | 54.2 | 65.3 | 69.0 |
| Sentiment140 | 64.9 | 64.4 | 70.6 | 59.2 |
| App2Check | 64.9 | 62.7 | 57.1 | 72.6 |
| SentiWordNet | 63.7 | 61.4 | 61.6 | 65.8 |
| Op. Lexicon | 62.5 | 50.8 | 65.3 | 59.8 |
| SOCAL | 62.1 | 50.9 | 65.5 | 58.8 |
| NRC Hashtag | 59.1 | 59.6 | 67.8 | 50.5 |
| H. Index | 53.9 | 44.8 | 45.9 | 61.8 |
| Stanford DL | 52.0 | 45.2 | 61.4 | 42.7 |
| Emolex | 51.1 | 45.8 | 61.4 | 40.7 |
| SASA | 47.1 | 37.9 | 44.6 | 49.7 |
| Op. Finder | 44.1 | 31.8 | 50.7 | 37.5 |
| Emoticons | 39.3 | 51.4 | 11.0 | 67.6 |
| Emoticon DS | 14.4 | 8.4 | 8.0 | 20.8 |
| Panast | 9.9 | 5.6 | 10.9 | 8.8 |
| SANN | 1.8 | 1.3 | 3.2 | 0.4 |

Table 7: Comparison on 1000 tweets in English

Hsinchun Chen. 2010. AI and opinion mining, part 2. *I.E.E.E. Computer Society*, pages 72–79.

Wei Gao and Fabrizio Sebastiani. 2015. Tweet sentiment: From classification to quantification. In *ASONAM*, pages 97–104. ACM.

Seth Grimes. 2010. Expert analysis: Is sentiment analysis an 80 http://www.informationweek.com/software/information-management/expert-analysis-is-sentiment-analysis-an-80–solution/d/d-id/1087919? [Online; accessed 5-May-2016].

Minqing Hu and Bing Liu. 2004. Mining and summarizing customer reviews. In *KDD*, pages 168–177. ACM.

Nuno Oliveira, Paulo Cortez, and Nelson Areal. 2013. On the predictability of stock market behavior using stocktwits sentiment and posting volume. In *EPIA*, volume 8154 of *Lecture Notes in Computer Science*, pages 355–365. Springer.

2016. Semeval-2016 task 4: Sentiment analysis in twitter.

Mike Thelwall, Kevan Buckley, Georgios Paltoglou, Di Cai, and Arvid Kappas. 2010. Sentiment strength detection in short informal text. *JASIST*, 61(12):2544–2558.

Andranik Tumasjan, Timm Oliver Sprenger, Philipp G. Sandner, and Isabell M. Welpe. 2010. Predicting elections with twitter: What 140 characters reveal about political sentiment. In *ICWSM*. The AAAI Press.

Theresa Wilson, Janyce Wiebe, and Paul Hoffmann. 2009. Recognizing contextual polarity: An exploration of features for phrase-level sentiment analysis. *Computational Linguistics*, 35(3):399–433.

