# Peer review of "App2Check and Tweet2Check: machine learning-based tools for Sentiment Analysis of Apps Reviews and Tweets"

_CoNLL 2016 — decision unknown_

[Official Review · Reviewer 1 · rating 1 · confidence 5]
soundness 2 · originality 2 · clarity 2 · impact 1 · substance 1 · appropriateness 2 · meaningful comparison 2 · replicability 1 · presentation format Poster

No details are provided on the methods used in this paper to produce the
results, due to issues of 'non-disclosure restrictions'.  If the reader doesn't
know the learning algorithm or the training data (or other resources made use
of in the approach), then there is nothing in the paper to help with the
reader's own sentiment analysis methods, which is why we share research.  This
is not a research paper, hence does not belong in this conference.  Perhaps a
submission to a demo session somewhere would be a good idea.  Even with a demo
paper, however, you would need to share more details about the methods used
than you do here.